# Offset Well Design Optimization Using a Surrogate Model and Metaheuristic Algorithms: A Bakken Case Study

**Ahmed Merzoug ***[ID] **and Vamegh Rasouli**

Department of Petroleum Engineering, University of North Dakota, Grand Forks, ND 58202, USA;
vamegh.rasouli@und.edu
* Correspondence: ahmed.merzoug@und.edu; Tel.: +1-701-215-5991

**Abstract:** Fracture-driven interaction FDI (colloquially called "Frac-hit") is the interference of fractures between two or more wells. This interference can have a significant impact on well production, depending on the unconventional play of interest (which can be positive or negative). In this work, the surrogate model was used along with metaheuristic optimization algorithms to optimize the completion design for a case study in the Bakken. A numerical model was built in a physics-based simulator that combines hydraulic fracturing, geomechanics, and reservoir numerical modeling as a continuous simulation. The stress was estimated using the anisotropic extended Eaton method. The fractures were calibrated using Microseismic Depletion Delineation (MDD) and microseismic events. The reservoir model was calibrated to 10 years of production data and bottom hole pressure by adjusting relative permeability curves. The stress changes due to depletion were calibrated using recorded pressure data from MDD and FDI. Once the model was calibrated, sensitivity analysis was run on the injected volumes, the number of clusters, the spacing between clusters, and the spacing between wells using Sobol and Latin Hypercube sampling. The results were used to build a surrogate model using an artificial neural network. The coefficient of correlation was in the order of 0.96 for both training and testing. The surrogate model was used to construct a net present value model for the whole system, which was then optimized using the Grey Wolf algorithm and the Particle Swarm Optimization algorithm, and the optimum design was reported. The optimum design is a combination of wider well spacing (1320 ft), tighter cluster spacing (22 ft), high injection volume (1950 STB/cluster), and a low cluster number per stage (seven clusters). This study suggests an optimum design for a horizontal well in the Bakken drilled next to a well that has been producing for ten years. The design can be deployed in new wells that are drilled next to depleted wells to optimize the system's oil production.

**Keywords:** surrogate modeling; optimization; hydraulic fracturing; fracture-driven interaction

## 1. Introduction

Hydraulic fracturing and horizontal drilling have unlocked huge amounts of oil and gas reserves. This technique allows operators to extract resources from low permeability rocks [1,2]. In the US, operators have been drilling wells on 640 acres to hold the lease before it expires in one to three years. This is called hold by production (HBP). Operators then come to perform infill drilling to increase production. The primary well is colloquially called "parent well", while the offset wells are called colloquially "child well". If the wells are drilled at the same time, they are colloquially called sibling wells. The problem in these configurations is the interference between the fractures of the offset wells and the primary well. This interference is called Fracture Driven Interaction, FDI, colloquially known as "Frac-Hit".

Ajani and Kelkar [3] and Daneshy et al. [4] were the first to present the concept of FDI. Several factors can promote the occurrence of FDI. These factors are reported in a thorough critical review by Gupta et al. [5]. Several statistical studies quantified this effect in terms

of productivity gains and production losses [6–8]. Table 1 summarizes the effect of FDI on production in several plays in the US.

**Table 1.** Effect of fractured child wells due to the production from the parent well (modified from Miller et al. [7]).

|  | **Bakken** | **Eagle Ford** | **Haynesville** | **Woodford** | **Niobrara** |
|---|---|---|---|---|---|
| Positive | 50% | 24% | 58% | 4% | 6% |
| No Change | 35% | 36% | 24% | 4% | 38% |
| Negative | 15% | 41% | 19% | 64% | 56% |

Cozby and Sharma [6] reported that the wells' spacing and produced volumes significantly affected the FDI intensity. They also reported that early wells in the Bakken exhibited positive behavior through FDI, mainly because these wells have been understimulated. This work mainly focused on the optimization of infill well completions that were drilled next to early wells in the Bakken.

Depletion is one of the many reasons that promotes the occurrence of FDI. Depletion is associated with poroelastic stress changes around the depleted primary well. This causes an asymmetric growth of the fractures from the offset wells towards the path of least resistance. The effect of depletion on FDI has been reported and studied by many researchers [3,9–13].

Many researchers have conducted numerical modeling studies to understand the effect of FDI on the performance of primary and offset wells. Kumar et al. [14] investigated the relationship between production losses, the diffusivity coefficient, and the well spacing. Rezaei et al. [15] simulated stress changes due to depletion and their effect on fracture propagation and reorientation from offset wells under different stress contrasts. Cipolla et al. [16] and Fowler et al. [9] modeled stress changes in a field-scale laboratory study. Morsy et al. [17] reported a numerical simulation workflow to optimize offset well design through sensitivity analysis in the Wolfcamp formation. Cai and Taleghani [18] worked on the theory developed by Daneshy [19] using numerical modeling for the interpretation of pressure response in primary wells during the stimulation of offset wells. Ratcliff et al. [13] modeled the increase in water production due to FDI in the Meramec of the STACK play. Their work modeled production losses and remediation techniques for conductivity damage.

Hydraulic Fracture (HF) treatment design optimization has been a topic of interest for both academia and the industry. Yu and Sepehrnoori [20] attempted to optimize the net present value (NPV) using numerical modeling and a response surface methodology (RSM) algorithm. They optimized the half-length, fracture spacing, and well spacing. Ma et al. [21] used reservoir simulation to optimize the design of HF stages. They used a covariance matrix adaptation evolution strategy, a genetic algorithm (GA), and simulations with perturbation stochastic approximation. Plaksina and Gildin [22] and Rahmanifard and Rahmanifard [23] attempted to optimize the number of stages, location, and fracture half-length using multi-objective optimization using numerical and analytical models. Cipolla et al. [24] used a fully integrated numerical workflow to optimize the Utica Shale's well spacing and perf cluster spacing. Kim and Choe [25] used proxy models through linear regression to optimize the HF design. The previous research was conducted using reservoir numerical simulation, and the science behind fracture propagation was ignored. Wang and Olson [26] developed a numerical three-dimensional fracture propagation code and optimized the fracture effective contact area using the generic algorithm (GA) and the pattern search algorithm (PSA). Their research focused only on the fracture contact area, which can sometimes be economically overstimulated. Garcia Ferrer et al. [27] suggested a workflow using numerical simulation to optimize child wells in a depleted environment. Wang et al. [28] used a data-driven approach in the Montney to maximize the productivity of child wells. Kang et al. [29] implemented a workflow for automated history matching and optimization for hydraulic fracturing design; in their study, they used a different optimization algorithm (COBYLA) and kriging as their proxy models.

Pudugramam et al. [30] optimized the well spacing and completion design for Hydraulic Fracturing Test Site 2 to maximize the net present value and return on investment.

In this work, a fully physics-based model was built and calibrated using data from logs, cores, microseismics, and production. The stress changes were estimated from and calibrated to microseismic depletion delineation (MDD). Once the model was built, a sensitivity analysis was run to build a proxy model. The simulation runs were structured using Sobol and Latin Hypercube sampling approaches. The proxy model was built using an artificial neural network (ANN) with an R_2 score of the order of 0.96. The proxy model was used to build a net present value function. Metaheuristic algorithms were used to optimize the net present value and find the optimum completion design for Bakken wells drilled next to a pre-depleted primary well.

## 2. Problem Statement

The setup of this project in the Bakken is part of a multi-year journey of data acquisition projects conducted by the operator to improve the reliability of the physics-based model. The project is called Red Sky (RS). The aim of the project was to understand well-to-well connectivity, drainage areas, and HF geometry. The setup is composed of a horizontal well (H1) and two vertical wells (V1 and V2) at a distance of 1000 ft and 1200 ft from the lateral of well H1 (see Figure 1). Table 2 summarizes the timeline for the setup.

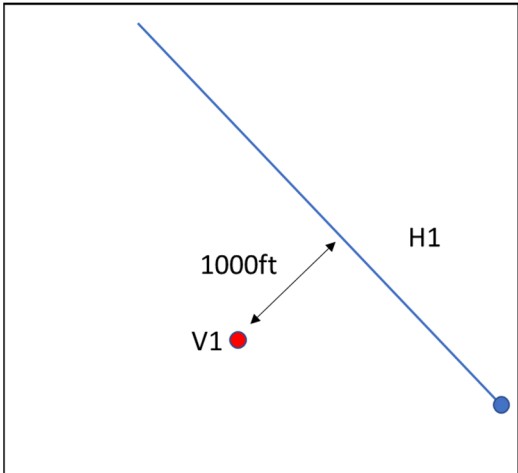

**Figure 1.** Schematic of the well location.

**Table 2.** Setup timeline.

| Production/Testing Event | Start Date | End Date |
| --- | --- | --- |
| Hydraulic Fracturing Treatment in H1 | 29 September 2005 | 29 September 2005 |
| H1 Primary Production | 29 September 2005 | 8 October 2016 |
| Shut-in H1 | 8 October 2016 | 5 January 2017 |
| V1 DFIT #1 | 7 November 2016 | 7 November 2016 |
| H1 MDD | 9 December 2016 | 11 December 2016 |
| V1 Fracture Stimulation | 11 December 2016 | 11 December 2016 |
| H1 Back to Production | 5 January 2017 | 3 May 2017 |

The completion design for H1 consists of one stage that was hydraulically fractured in one go using ball sealers and six diversion stages. The stage consists of 33 clusters separated by 134 ft. The pumping treatment had a very small total injected volume of 4600 bbl of XL fluid pumped at a rate of 60 bpm and a total of 420,000 lb of 20/40 sand. The radioactive

(RA) tracer showed very good coverage of proppant in the 33 clusters. The well produced 110,000 bbl prior to microseismic depletion delineation (MDD).

The MDD was first suggested by Dohmen et al. [31]. It was mainly used to map the depleted area (an approximation to fracture geometry) and measure stress changes. More details about the interpretation can be found in Cipolla et al. [16].

The time-lapse DFITs were interpreted by Cipolla et al. [16] using the compliance method with an estimated closure stress of 7150 psi and a pore pressure of 5800 psi in the Middle Bakken Formation. Note that in the model calibration, only the MDD, DFIT #1, and V1 fracture treatment responses were matched. The reason for doing this is that calibrating the pressure response from the MDD is sufficient for stress change calibration because it represents an average value for all H1. In contrast, the DFITs represent local measurements that the heterogeneity of the formation can cause. Another difference between this work and the work of Cipolla et al. [16] and Fowler et al. [9] is that the contribution of the Three Forks is considered in the modeling. The numerical modeling section will discuss more details about the modeling process.

### 3. Numerical Modeling

The numerical model for this study was built based on data from Cipolla et al. [16] and Fowler et al. [9] with some minor changes. The numerical simulation was run in a fully integrated physics-based simulator. The numerical simulator incorporates modeling hydraulic fracturing with reservoir and geomechanics components integrated. This approach was adopted to build a representative model of hydraulic fracture propagation and calibrate the stress changes due to depletion and reservoir performance. The numerical simulator used has independent meshes between the fracture and the reservoir grid. The two components are connected using a 1D submesh to capture the fluid flow between the rock matrix grid and the fracture grid. This approach overcomes previously used approaches because it couples most of the physics in the system without ignoring parameters when going from hydraulic fracturing simulation to reservoir simulation. The numerical model provides more confidence in the generated results because all aspects have been constrained in one way or another. Figure 2 illustrates the workflow used for building the numerical model.

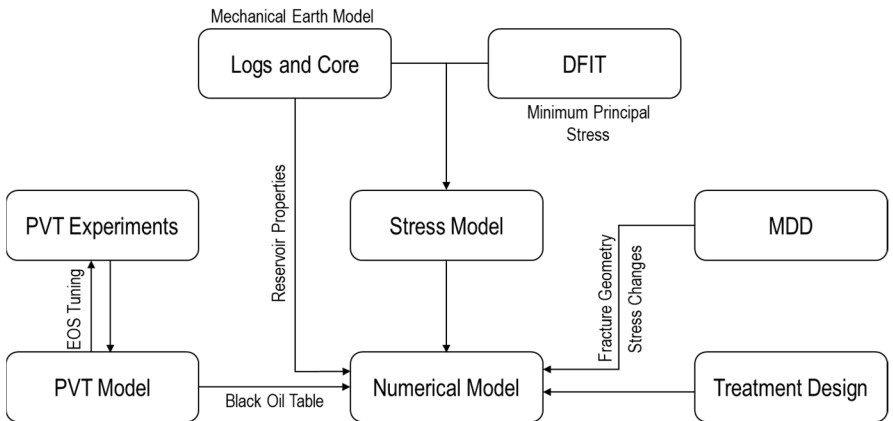

**Figure 2.** Workflow for building a numerical model.

The stress model was estimated using the transverse isotropic (TI) stress model (see Appendix A). The Diagnostic Fracture Injection Tests (DFIT) were interpreted using the compliance method as reported in Cipolla et al. [16] and Fowler et al. [9]. The MDD, along with radioactive tracer data, were used to calibrate fracture geometry. The MDD pressure response was used to calibrate stress changes in the formation. The PVT model was calibrated using Constant Composition Expansion and Differential Liberation data in a PVT numerical simulator. A black oil PVT was generated and used as an input for the numerical model. Reservoir and rock properties were taken from data reported by Cipolla et al. [16].

The fracture geometry from the horizontal well was estimated from MDD. The mapping results show events from the lower Three Forks to the Lodgepole with a height growth of approximately 250 ft. The lateral growth is averaged at 500 ft, with a dominant fracture at the heel of the well at 1000 ft. The recorded microseismic events belong only to the first 2000 ft of the lateral. This suggests that the toe of the well was not drained effectively. The drainage can be explained by pressure losses along the lateral, which minimize drainage at the toe of the well. The low drainage did not alter the stress state at the toe of the well. This is supported by the effectiveness of the treatment shown in the radioactive tracers. Figure 3 illustrates the recorded microseismic events. In the modeling process, both RA and microseismic data were combined to model and calibrate the lateral extent of the fractures using a fracture toughness of 2000 psi·in$^{0.5}$. The height was estimated and calibrated by building a mechanical earth model (MEM) and a DFIT (see Appendix A).

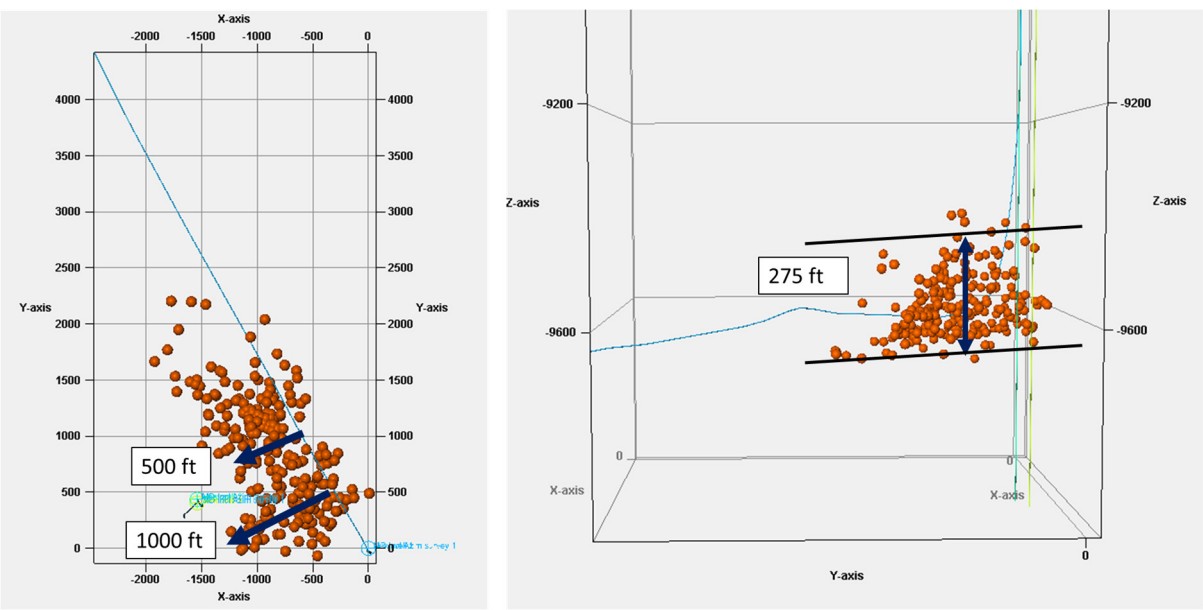

**Figure 3.** Microseismic Depletion Delineation recorded event and estimated fracture geometry.

The geomodel was built using data from logs, cores, and data reported by Cipolla et al. [16]. The rock properties are summarized in Table 3. The production is assumed to be only from the middle Bakken (MB) (divided into four by Cipolla et al. [16]) and Three Forks (TF) Formations, whereas the permeability in the Lodgepole, Upper, and Lower Bakken Shales was assumed to be zero [32,33]. The production history was matched by adjusting the matrix relative permeability curves using a total liquid production constraint to ensure mass conservation. Figure 4 reports the adjusted relative permeability curves from Cho et al. [34] and Cipolla et al. [16]. The history matching results are reported in Figure 5. The grid sizes are reported as follows:

- A fracture grid size of 80 ft was used. This value is acceptable for the range of application of the fracture propagation algorithm [35];
- A geomodel with 5635 ft height, 15,000 ft length, and 1280 ft height was built;
- A logarithmic grid length in the Shmax direction was used to account for the sensitivity analysis fracture geometry.

**Table 3.** Reservoir rock properties used in the geomodel.

| Formation | Permeability | Porosity | Water Saturation |
|-----------|--------------|----------|------------------|
| MB1 | 0.004 | 6.5 | 42 |
| MB2 | 0.0026 | 7 | 36 |
| MB3 | 0.0026 | 5.5 | 42 |
| MB4 | 0.001 | 5.5 | 40 |
| TF | 0.0015 | 5 | 40 |

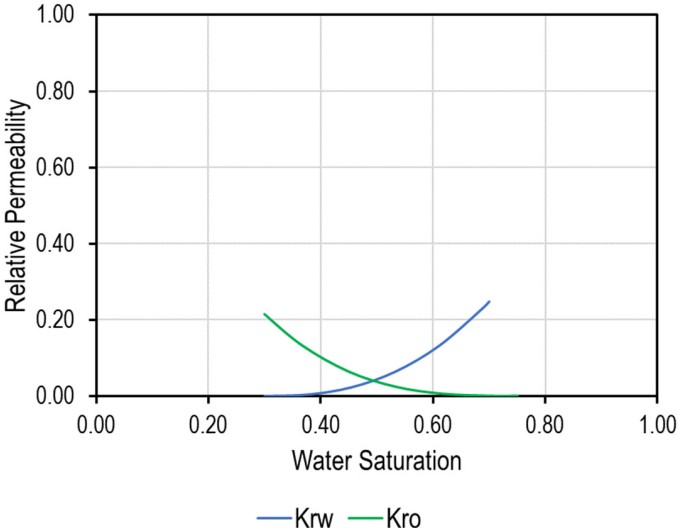

**Figure 4.** Relative permeability curves used in the history match.

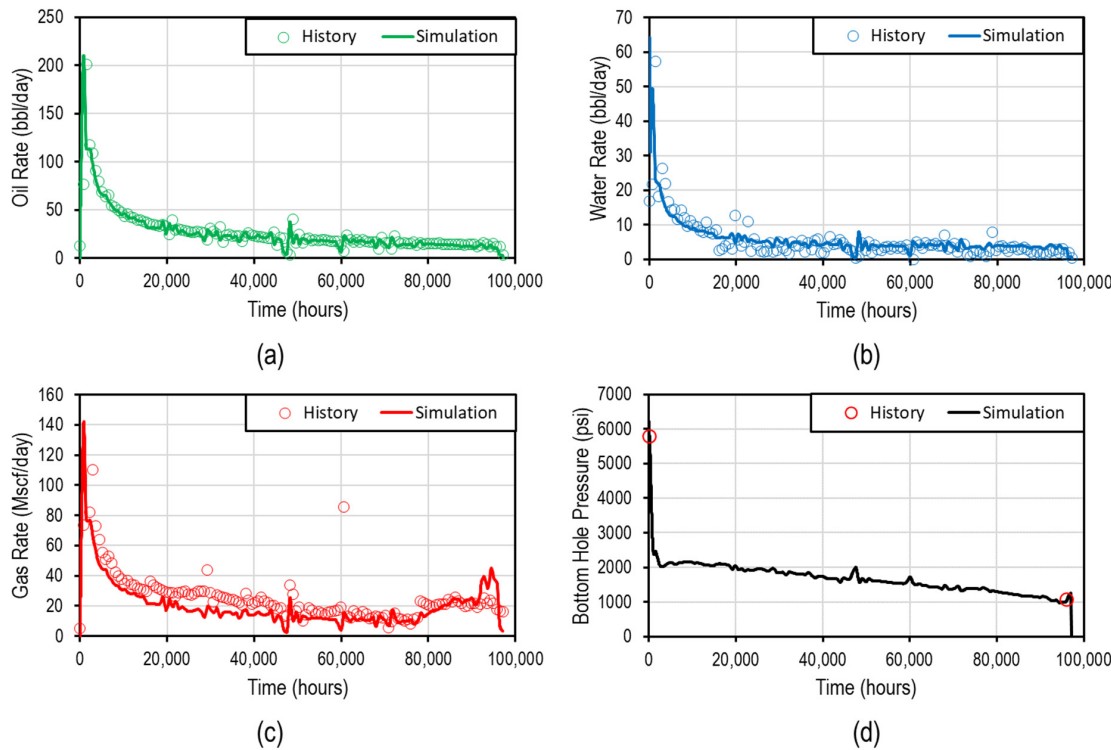

**Figure 5.** Production and bottom-hole pressure match. (**a**) oil rate match, (**b**) water rate match, (**c**) gas rate match, and (**d**) bottom hole pressure match.

Stress changes due to depletion were measured through MDD and hydraulic fracturing treatment at well V1. The pressure of the MDD stabilizes at 6000 psi. This pressure is close to the magnitude of FDI, with a magnitude of 5100 psi. The magnitude of the FDI pressure is related to the location of the occurrence of the fracture interaction. Communication can take several forms. Direct connection between fractures from the offset well and the primary well, drainage area overlap, and connection through fractures [5]. Daneshy [36] laid the groundwork for the interpretation of the pressure response; however, the interpretation process is still subjective, and the explanations can vary. According to the theory of poroelasticity, stress changes are dependent on the Biot coefficient and the poroelasticity coefficient [37]. The stress changes can be estimated using the following equation [37].

$$\Delta S_{hmin} = \alpha \left( \frac{1 - 2\nu}{1 - \nu} \right) \tag{1}$$

To match the pressure change, the Biot coefficient was adjusted following the process described in Fowler et al. [9]. The calibration value of the Biot coefficient was 0.7. This value is different from the study by Fowler et al. [9] for the same case study, where they reported a value of 0.34. The discrepancy in the results is due to the different approaches used for modeling fractures in well H1. In their work, the approach was to implement pre-existing fractures that force the simulator to have uniform drainage along the fractures. In the current work, the fractures were modeled by creating lower drainage towards the tip of the fracture. These effects change the MDD modeling response in the two cases. Figure 6 illustrates the bottom hole pressure match for the MDD part and the pressure response due to the fracture interaction from the offset well V1. The Biot coefficient of 0.7 was sufficient to match both fracture interaction and the MDD. This match gives more confidence to our model in terms of poroelastic stress change. Figure 7 illustrates the poroelastic stress changes and the FDI from well V1. Note that the fracture from well V1 is connected to the fracture from the horizontal well.

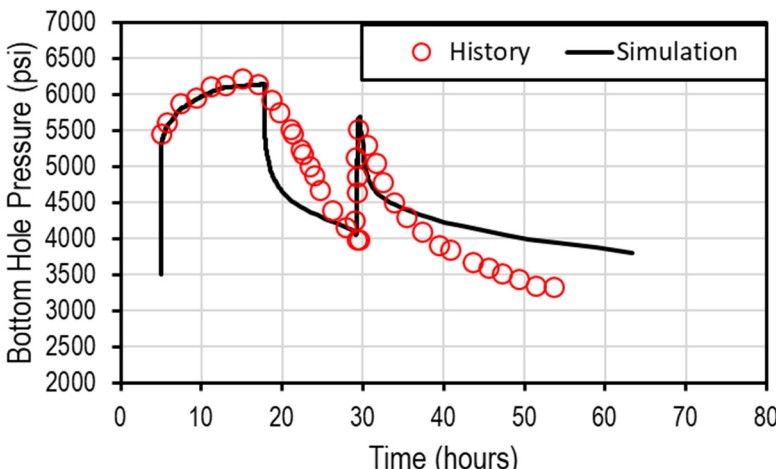

**Figure 6.** MDD pressure response and fracture-driven interaction match.

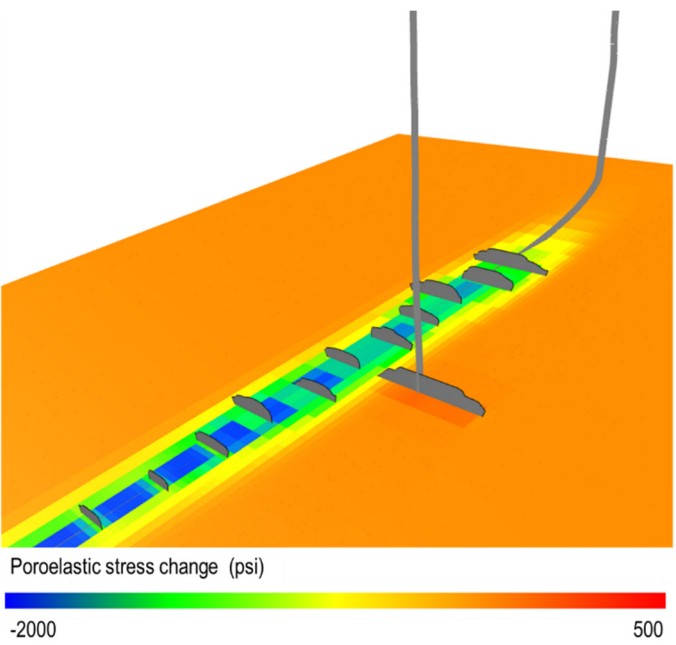

**Figure 7.** Predicted poroelastic stress changes and fracture driven interaction.

### 4. Optimization Formulation

The optimization process requires a mathematical formulation for the problem's so-called objective function. This step can be addressed by building a proxy model. Proxy models are objective-oriented; thus, a good understanding of the problem and control variables is necessary before embarking on the modeling process. In this work, the objective function is the net present value (NPV). It is expressed as follows:

$$\text{NPV} = f(IV, CN, CS, WS) \times O_p - SV_p \times IV \times TCN - NPT_C \times SN \tag{2}$$

where:

$IV$: Injected volume per cluster;
$CN$: Number of clusters per stage;
$CS$: Cluster spacing;
$WS$: Well spacing;
$O_p$: Oil price;
$SV_p$: Fracture volume price per barrel;
$TCN$: Total clusters number;
$NPT_C$: Non-productive time cost (time between stages);
$SN$: Stages number.

The function $f(IV, CN, CS, WS)$ was built through proxy modeling using neural networks. The ANN model was used because the problem is an interpolation one. Thus, it would be fine to overfit the parameters in the solution domain. The design parameters and their range are summarized in Table 4.

To ensure that the pumping schedule used is realistic, the treatment design was obtained from Cipolla et al. [38]. The design was for 14 clusters (with two perforations each) in one stage with a total injected volume of 7000 bbl. This treatment was upscaled to the sensitivity analysis intended by keeping the pumping rates and the proppant concentrations constant and only changing the time to adjust the total volume of injected fluid. This approach will automatically upscale the total volume and the injected proppant. Figure 8 illustrates the treatment design pumped in the Bakken. The scaling process was calculated as follows:

$$ratio = \frac{IV \times CN}{7000} \tag{3}$$

The ratio is then multiplied by the time intervals of the design depicted in Figure 8 to scale the total volume of fluid and total proppant.

**Table 4.** Design parameters for the hydraulic fracturing job.

| Parameter | Unit | Range |
|---|---|---|
| Injected volume per cluster | bbl/cluster | 400–2000 |
| Number of clusters | / | 7–20 |
| Spacing between the clusters | ft | 15–50 |
| Well spacing | ft | 440; 660; 880; 1320 |
| Treatment design | | High-Viscosity Friction Reducer (Proppant size: 50% 100 mesh and 50% 40/70 mesh) |

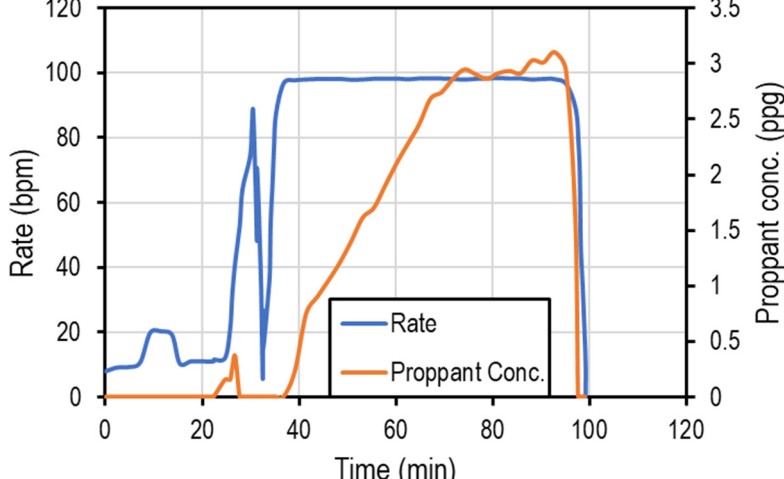

**Figure 8.** Hydraulic fracturing treatment schedule for one stage.

The neural network used had an architecture of seven layers. The input layer has four inputs, followed by five hidden layers, the first three of which each have six neurons. The fourth layer had four neurons, followed by a three neuron layer, and finally an output layer. The activation function was ReLu, and the optimizer was Adams. A total of 1000 iterations were performed during the training. These parameters were selected based on trial and error.

Before any training, the data was normalized as follows to reduce the space of solutions for the Adam optimizer [32]. The following equation was used:

$$X_{Normalized} = \frac{X - \mu}{\sigma} \tag{4}$$

where $X_{Normalized}$ is the normalized data, $X$ is the data, $\mu$ is the average, and $\sigma$ is the standard deviation.

The error was quantified using the root mean square error (RMSE) as follows:

$$\text{RMSE} = \sqrt{\frac{\sum_{i=1}^{n} \left(Y_i^{proxy} - Y_i^{simulation}\right)^2}{n}} \tag{5}$$

where $n$ is the number of scenarios and $Y_i$ is the prediction result for the proxy model or simulation.

Two metaheuristic algorithms were used to optimize the completion design in this study. These algorithms were selected because they are gradient-free algorithms. The advantage of incorporating these algorithms into an optimization process is that they

help to find a global optimum due to their stochastic search patterns, which prevent local optimum issues [39]. Both algorithms are discussed in the following section.

*4.1. Grey Wolf Optimization Algorithm*

Mirjalili et al. [40] introduced the algorithm Grey Wolf Optimization, which is inspired by the Grey Wolf's behavior in nature. The algorithm has been applied in the petroleum engineering literature [40–44].

The wolves act in a hierarchical order. First, the alpha wolf is the leader of the pack. The alpha wolf is responsible for making decisions about hunting prey. Second, the beta wolves are subordinate to the alpha wolves in making and enforcing the decision. These wolves are the leaders of the pack. Next, the delta wolves are subordinate to the alpha and beta wolves. The lowest level of the hierarchy are the omega wolves [40].

The hunting process follows the hierarchy of the wolves. This process was modeled mathematically as the Grey Wolf algorithm to optimize processes and objective functions. The algorithm proceeds as follows [41]:

- Generate a random set of solutions (can be bounded) to represent the location of wolves in the solution space;
- Evaluate the locations according to the cost function (NPV for this work);
- Rank the solutions and assign them according to the hierarchy of the wolves $\alpha$, $\beta$, $\delta$, and the rest of the solutions to $\gamma$;
- Update the location of the wolves according to the best solution (assumed to be alpha);
- Repeat the process until the maximum number of iterations is reached.

*4.2. Particle Swarm Optimization Algorithm*

The Particle Swarm Optimization algorithm mimics the natural behavior of insects, birds, and fish. It was first suggested by Kennedy and Eberhart [45]. Each solution represents the position of a particle. The positions of each particle are updated from one iteration to another. These solutions are characterized by a position and a velocity [46]. Three parameters mainly control the algorithm [47]:

- *Cognitive*: serves as the memory of the particle; ensures that the particle moves towards the best values; and limits the step size in the search and convergence process;
- *Social*: determines the step size while converging to the swarm's best solution;
- *Inertia*: control the speed of convergence and encourage exploration of new solutions.

The algorithm is executed as follows [48]:

- Initialize the PSO parameters;
- Generate primary swarm positions;
- Evaluate the fitness of each position using the objective function (NPV);
- Record the best position for each particle along with the global best particle;
- Update the position and the velocity of the particles until the maximum number of iterations is reached.

## 5. Sensitivity Analysis

Different sampling approaches have been used to reduce the total number of simulations needed to build a proxy model. Eighty different treatment designs were generated using Latin Hypercube sampling [49] and Sobol Sequence sampling [50]. The combination of these approaches allows the generation of spaced data points that efficiently represent the space of solutions [51]. The computation is run on the cloud. A restart file was used to reduce the computation time. Modeling multistage hydraulic fracturing and production for all the wells can be time-consuming; thus, only one stage of the well in each scenario was simulated. The results were then upscaled to the full well, assuming each stage has an equal contribution to the total production of the well.

Note that local grid refinement was used at the level of the stage. The total length of the refined grid is 1000 ft, which is equal to the longest possible stage with 20 clusters of

50 ft spacing. The grid was refined to 15 ft, which is the smallest possible spacing to ensure that for all cases of sensitivity, one fracture exists per grid [52].

The results were quantified for three and ten year production as follows:

- Offset well by cumulative production normalized by length;
- Primary well cumulative production uplift normalized by length.

The sum of the two quantities represents system production normalized by length. Figure 9 illustrates a simulation case for the sensitivity analysis.

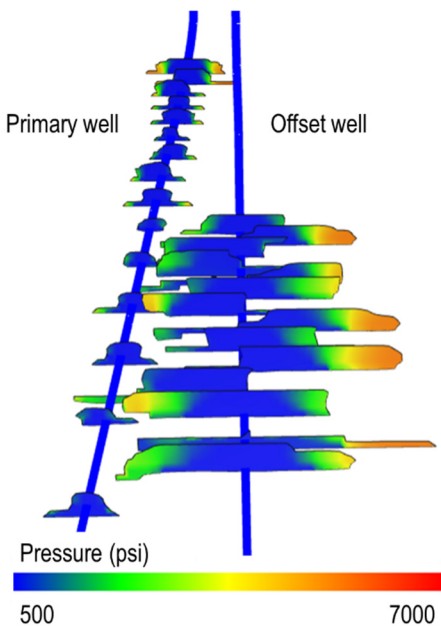

**Figure 9.** Sensitivity analysis case for one-stage simulation.

## 6. Results and Discussion

During the sensitivity analysis, there were cases with a tighter well spacing (440 ft) that reached the maximum surface pressure constraint. This is because of high-stress shadow values between clusters. The high-stress shadow is a result of the low stress because of the depletion near the primary well. This causes the net pressure to increase, leading to a higher width, which causes high stress values.

The neural network was trained using 90 data points that were generated using sampling approaches. The model was then incorporated into the NPV function and optimized. Ten simulations were run around the optimum design to increase the accuracy of the results and the model. The neural network training and testing results are illustrated in Figure 10 with a R_score of 0.961 and 0.9678 for the training and the testing, respectively. The RMSE is reported as 4.03 and 3.18 for the training and testing, respectively, as can be seen from Figure 10, which shows the acceptable error between the simulation and the proxy model. The simulations were run on the cloud using a high-performance computer from Microsoft Azure. The minimum simulation time for one case is 20 h, whereas the proxy model can give acceptable results in less than a second.

The NPV function is calculated using the following assumptions:

- Oil price $(O_p)$ USD 80;
- Slurry price $(SV_p)$ USD 100/bbl;
- The cost of adding a new stage $(NPT_C)$ USD 2000.

The neural network was used to run sensitivity analysis and understand the effect of the offset well completion on the performance of the system. Figure 11a illustrates the effect of well spacing and cluster spacing on the NPV. The assumption is that 1000 bbl were injected per cluster, with seven clusters per stage. It can be noted that wider spacing

results in better performance. This is attributed to the fact that wider spacing results in a larger drainage volume. The optimum number of clusters is around 22 for all well-spaced cases. Figure 11b illustrates the effect of injected volume and cluster spacing on the NPV. The assumption is that the well has seven stages and 1320 ft spacing between wells. It can be noted that, overall, larger volumes result in a better NPV. However, the optimum cluster spacing was wider for bigger injection volumes. This is because larger injection volumes result in longer fractures, implying larger drainage volumes that cause faster interference. Figure 11c illustrates the effect of cluster number per stage and cluster spacing. The assumption is that the injected volume is 1000 bbl per cluster and the well spacing is 1320 ft. It is observed that fewer clusters per stage result in better performance. This was because the fluid velocity at lower cluster numbers per stage was higher per fracture (limited maximum injection rate). The higher velocity results in better proppant placement in the fracture. The optimum cluster spacing is around 20 ft.

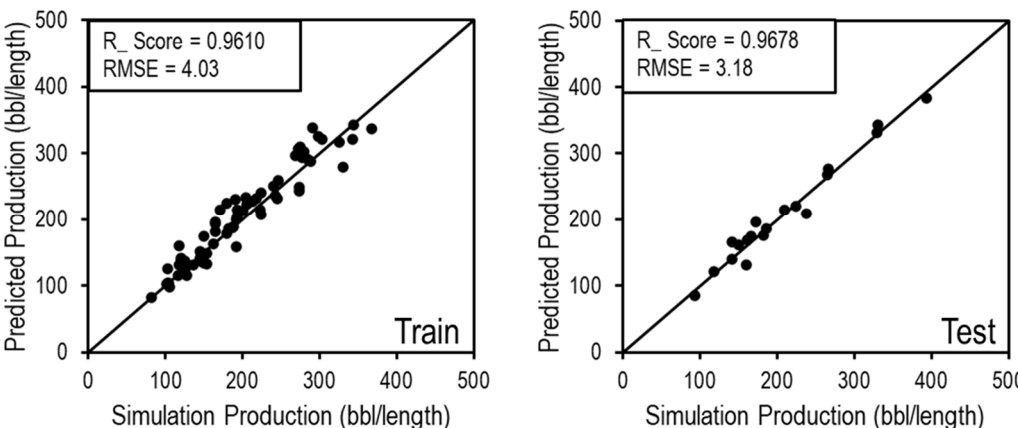

**Figure 10.** Results of training and testing neural networks.

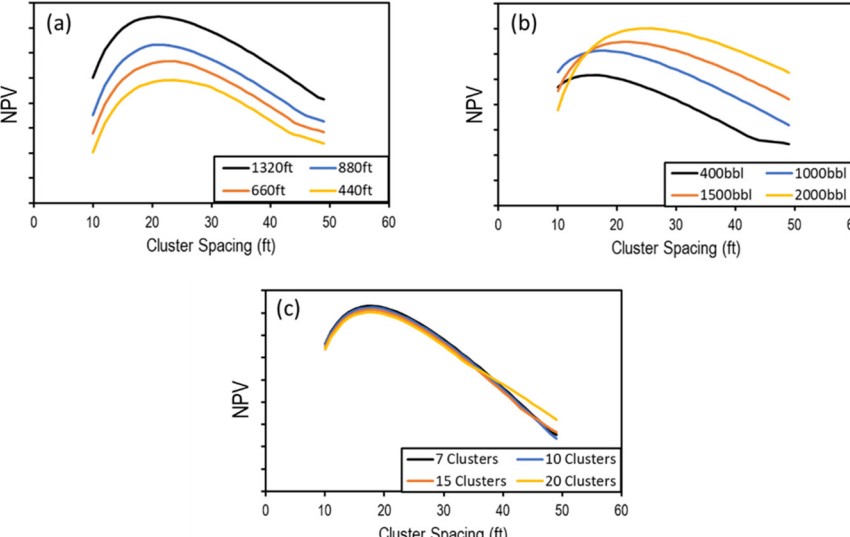

**Figure 11.** (**a**) The effect of well spacing and cluster spacing on NPV, (**b**) the effect of injected volume per cluster and cluster spacing on NPV, and (**c**) the effect of the number of clusters per stage and the cluster spacing on NPV.

The optimization algorithms were run on the NPV. The algorithms used libraries for the Grey Wolf algorithm [53], and Particle Swarm Optimization [54]. Note that the results reported by these algorithms depend on the economic assumptions of the oil price and the completion cost. Table 5 reports the optimization results. Note that these values assume all clusters contribute equally to production.

**Table 5.** Optimization results for NPV using different optimization algorithms.

| Algorithm | Injected Volume (bbl/Cluster) | Cluster Number | Cluster Spacing (ft) | Well Spacing (ft) | Error from Sim % |
|---|---|---|---|---|---|
| GWO | 1950 | 7 | 25 | 1320 | 1% |
| PSO | 1900 | 7 | 26 | 1320 | 1.2% |

### 7. Conclusions

In this study, we used a surrogate model to understand the effect of stimulation design parameters on the net present value in the Bakken system. We successfully represented the system using a neural network based on a physics-based, fully integrated simulator that accounts for fracture propagation, reservoir fluid flow, and stress changes due to geomechanics. The model was calibrated using different sources of hard data, which gives more confidence in the results. The design was then optimized using metaheuristic algorithms (Grey Wolf Optimization and Particle Swarm Optimization). The numerical simulation was time-consuming, with a total of around 4000 h running on the cloud. The use of a surrogate model reduced the simulation time and provided a proxy for system performance. The model suggests that the optimum design had wider well spacing (1320 ft), tight cluster spacing (22 ft), a low number of clusters per stage (seven clusters), and a large injection volume (1950 STB). The proxy model served the purpose of optimization well and was confirmed by the simulation. Since it has potential, this workflow can be applied to newly drilled wells and for refracturing.

**Author Contributions:** Conceptualization, A.M.; Methodology, A.M.; Software, A.M.; Validation, A.M.; Formal analysis, A.M.; Writing—original draft, A.M.; Writing—review & editing, V.R.; Supervision, V.R.; Funding acquisition, V.R. All authors have read and agreed to the published version of the manuscript.

**Funding:** This research was possible through North Dakota Industrial Commission (NDIC) for the financial support contract NDIC G-045-89.

**Data Availability Statement:** Data available upon request.

**Acknowledgments:** The authors would like to acknowledge the support from the Hess Corporation for providing the data and validating the results. We appreciate the ResFrac team for their software support and cloud computing power. We would also like to thank Neal Nagel and Marisela Sanchez-Nagel for reviewing the paper.

**Conflicts of Interest:** The authors declare no conflict of interest.

### Appendix A

The mechanical earth model was built using the anisotropic extended Eaton method for transverse isotropic media [55]. The approach assumes that the overburden's stress is applied instantaneously to an elastic rock. The horizontal stresses are a result of tectonic stress and overburden. The approach has several degrees of freedom [56]. In this work, freedom was reduced by using different data sources. The stress was estimated from well logs and calibrated with core data and DFIT values. The used DFIT values are reported in the work of Dohmen et al. [57] and Cipolla et al. [16]. Figure A1 illustrates the calibrated mechanical earth model.

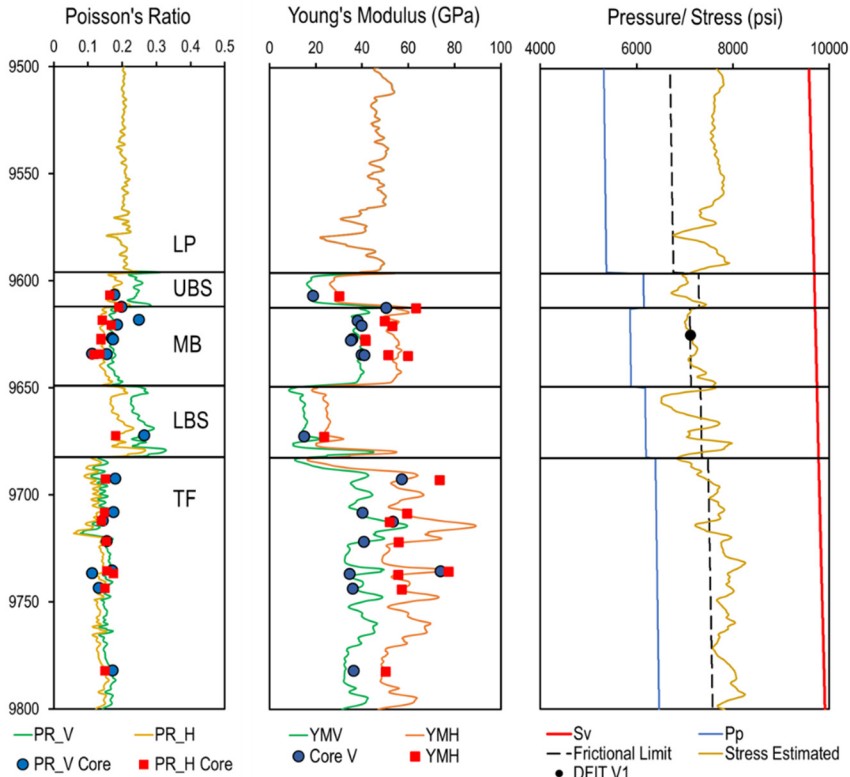

**Figure A1.** From left to right: Poisson's ratio, Young's modulus, stresses, and pressure. The lines are data predicted from logs and calibrated. The points represent measured core data. V stands for vertical, and H stands for horizontal.

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
