# Peer review of "Offset Well Design Optimization Using a Surrogate Model and Metaheuristic Algorithms: A Bakken Case Study"

_2673-4117, doi:10.3390/eng4020075_

Round 1
Reviewer 1 Report
Appendix A can be introduced within the normal article text.
The abstract and conclusion is not detailing the results, and analysis.
It is very important to address some quantitative information in the abstract.
It is very important to address some quantitative information in the conclusion.
It is important to detail better the data used (where it comes from, who provided, approvals, etc.).
The authors use most of the time recent articles as a reference, what is very positive.
The text is good structured and organized.
Author Response
Thank you so much for the time to review the manuscript. I really appreciate it.
Appendix A can be introduced within the normal article text.
We have put it in Appendix because it was not part of the main work. We think that keeping it in the Appendix would make more sense.
The abstract and conclusion is not detailing the results, and analysis.
It is very important to address some quantitative information in the abstract.
It is very important to address some quantitative information in the conclusion.
Indeed the two components have been adressed
It is important to detail better the data used (where it comes from, who provided, approvals, etc.).
The data was obtained from Cipolla et al. 2020. the reader is referred to check more detail about the data on that paper
Thank you so much
Reviewer 2 Report
This manuscript proposes a fully physics-based model and calibrates using data from logs, cores, microseismic, and production. The work is overall logically structured and the present with good technical details. This paper makes a good contribution to the oil production. There are several places in this manuscript requiring further clarification and modifications as suggested below. A minor revision is required.
1. The English needs to be improved.
2. The author should add some recent research literature to the part of the numerical simulation in Section Introduction, such as “Meng et al. (2023) Rigid-Block DEM Modeling of Mesoscale Fracture Behavior of Concrete with Random Aggregates. Journal of Engineering Mechanics. Doi: 10.1061/JENMDT.EMENG-6784.” and “Meng et al. (2020) Three-dimensional mesoscale computational modeling of soil-rock mixtures with concave particles. Engineering Geology, 277, Doi: 10.1016/j.enggeo.2020.105802.”.
3. Why are there two Table 2? Please ask the author to re-label the Table.
4. The format of the full paper should be unified. For example, Fig. 1 has a bold font, while Fig. 2 has no bold font in the paper.
5. What are the advantages and disadvantages of the numerical model in this paper compared to traditional models?It is recommended that the author create a Table to help the reader understand.
6. The equations in Section “Optimization Formulation” and subsequent sections are not numbered. In addition, the multiplication sign is generally used “×” instead of “*” in the equations.
7. The introduction of the two metaheuristic algorithms is recommended to be merged into a single section. The structure of the article seems cluttered.
8. Please list the advantages of Grey Wolf Optimization Algorithm and Particle Swarm Optimization Algorithm. Considering that there are more than just these two gradient free algorithms, why did the author choose these two algorithms?
9. “The ratio is then multiplied by the time intervals of the design depicted in Fig to scale
the total volume of fluid and total proppant.” This sentence should indicate the No. of figure to avoid misunderstanding.
10. In civil engineering related writing, the first person is generally avoided, such as “In this study, We used a surrogate model to understand the effect of stimulation design parameters on the net present value in the Bakken system.”. In addition, the “W” in “We” is a lower case.
Author Response
Thank you so much for reviewing the paper. I really appreciate your time.
- The English needs to be improved.
English was reviewed. Thank you so much
- Why are there two Table 2? Please ask the author to re-label the Table.
Fixed thank you
- The format of the full paper should be unified. For example, Fig. 1 has a bold font, while Fig. 2 has no bold font in the paper.
Fixed Thank you
- What are the advantages and disadvantages of the numerical model in this paper compared to traditional models?It is recommended that the author create a Table to help the reader understand.
A paragraph was added. a comparison with other models would need to review several other simulators such as CMG, Gohfer where every one of them has different approaches to solve the problem. THus we only focused on the special thing about the simulator we used
- The equations in Section “Optimization Formulation” and subsequent sections are not numbered. In addition, the multiplication sign is generally used “×” instead of “*” in the equations.
Fixed Thank you
- The introduction of the two metaheuristic algorithms is recommended to be merged into a single section. The structure of the article seems cluttered.
Fixed Thank you
- Please list the advantages of Grey Wolf Optimization Algorithm and Particle Swarm Optimization Algorithm. Considering that there are more than just these two gradient free algorithms, why did the author choose these two algorithms?
We used the algorithms because we are comfortable using them. we expect other algorithms to behave similarly. we also compare between these two algorithms
- “The ratio is then multiplied by the time intervals of the design depicted in Fig to scale
the total volume of fluid and total proppant.” This sentence should indicate the No. of figure to avoid misunderstanding.
Fixed THank you
- In civil engineering related writing, the first person is generally avoided, such as “In this study, We used a surrogate model to understand the effect of stimulation design parameters on the net present value in the Bakken system.”.In addition, the “W” in “We” is a lower case.
Fixed Thanks
Round 2
Reviewer 1 Report
Already presented previously.